# The Digital Pharmacies Era: How 3D Printing Technology Using Fused Deposition Modeling Can Become a Reality

**DOI:** 10.3390/pharmaceutics11030128

**Published:** 2019-03-19

**Authors:** Maisa R. P. Araújo, Livia L. Sa-Barreto, Tais Gratieri, Guilherme M. Gelfuso, Marcilio Cunha-Filho

**Affiliations:** Laboratory of Food, Drugs and Cosmetics (LTMAC), University of Brasília (UnB), Brasília 70910-900, Brazil; maisaraposo@gmail.com (M.R.P.A.); liviabarreto@unb.br (L.L.S.-B.); tgratieri@gmail.com (T.G.); gmgelfuso@unb.br (G.M.G.)

**Keywords:** digital pharmacy, fused deposition modeling 3D printing, modified drug release, personalized medicines, telemedicine

## Abstract

The pharmaceutical industry is set to join the fourth industrial revolution with the 3D printing of medicines. The application of 3D printers in compounding pharmacies will turn them into digital pharmacies, wrapping up the telemedicine care cycle and definitively modifying the pharmacotherapeutic treatment of patients. Fused deposition modeling 3D printing technology melts extruded drug-loaded filaments into any dosage form; and allows the obtainment of flexible dosages with different shapes, multiple active pharmaceutical ingredients and modulated drug release kinetics—in other words, offering customized medicine. This work aimed to present an update on this technology, discussing its challenges. The co-participation of the pharmaceutical industry and compounding pharmacies seems to be the best way to turn this technology into reality. The pharmaceutical industry can produce drug-loaded filaments on a large scale with the necessary quality and safety guarantees; while digital pharmacies can transform the filaments into personalized medicine according to specific prescriptions. For this to occur, adaptations in commercial 3D printers will need to meet health requirements for drug products preparation, and it will be necessary to make advances in regulatory gaps and discussions on patent protection. Thus, despite the conservatism of the sector, 3D drug printing has the potential to become the biggest technological leap ever seen in the pharmaceutical segment, and according to the most optimistic prognostics, it will soon be within reach.

## 1. Introduction

The industrial revolution in its beginning transformed drug therapy with large-scale production in assembly line, symbolized by the production of tablets in 1834. Thenceforward, despite the modernization of industrial facilities and advances in quality issues, the bases of the pharmaceutical production process were not modified. 3D printing has revolutionized various sectors of human activity over the last few decades, being one of the pillars of the fourth industrial revolution. In recent years, the use of this technology in medicine preparation has demonstrated such potential. This is why experts around the world point out that the pharmaceutical field has finally been given, after two centuries, the opportunity to make a significant technological jump [1].

3D technology offers unique benefits to drug products manufacturing, when compared to traditional methods—notably, the capacity of designing personalized pharmaceutical forms with flexible dosage [2,3,4,5], different shapes [3,6,7], multiple active pharmaceutical ingredients (even incompatible ones) [8,9], and modulated release kinetics [10,11,12,13]. Moreover, the most diversified and sophisticated drug delivery devices for oral, dermal, and implantable administration can be produced with high accuracy using 3D printers [14].

Nonetheless, competing against the mass production of pharmaceutical manufacturing is not a simple task. The FDM 3D printer may not be an optimal solution for largescale production. For instance, a tablet machine can be 60 times faster than a printer [15]. 3D printers cannot match the velocity of industrial tablet machines, but they certainly serve to address an existing therapeutic gap with regard to the need for individualization of drug therapy, acting in a complementary or alternative manner to the conventional drug product production [16,17].

On the other hand, in a small scale at compounding pharmacies, the speed difference between manually encapsulating powders and using a printer may not be so discrepant. The printer can also be adapted with multiple heads, making it possible to print several units at the same time, accelerating the process. The artisanal processes employed today at compounding pharmacies to meet patient’s individuals needs are similar to those used in apothecaries hundreds of years ago. For instance, they are not capable of elaborating controlled-release dosage forms and are not sufficiently equipped to guarantee the quality specifications required by the pharmacopoeia, risking patient safety [18]. The automation of the 3D printing process and particularly the high accuracy achieved by FDM technology makes the printing of drug products potentially safer. It could also prevent low-quality issues and meet the necessary requirements to take drug personalization to another level [14,15,19,20,21].

Fused deposition modeling (FDM), a type of 3D printing technology, is the most quoted when dealing with production of drug delivery devices, because of the low cost of printers; printing precision, fundamental to guaranteeing medicine quality parameters; and hot-melt extrusion, a technological process incorporated in the pharmaceutical field a decade ago [22,23] The FDM 3D printer uses heat to melt a polymeric filament and deposit it layer by layer in the *x*, *y* and *z*-axes, creating a three-dimensional product [24,25]. The filament used to feed the printer is produced by hot-melt extrusion using active pharmaceutical ingredients and pharmaceutical grade polymers [26].

The co-participation of both pharmaceutical industry and compounding pharmacy seems to be the best way to cross the barrier of research and reach the market. The FDM 3D printer is portable and relatively simple to operate, making it eligible for implementation in compounding pharmacies. On the other hand, the hot-melt extruded drug loaded filaments can be produced by the industry on a large scale as an intermediate product. These filaments can be transformed into personalized medicines for medical prescriptions at local pharmacies [27].

Moreover, personalized 3D printed medicines appear as the missing piece in the care cycle of trendy telemedicine (Figure 1). Launched as the future of medicine, telemedicine has the capacity to expand access to health, making it possible to contact patients from the neediest regions of the globe with the most qualified physicians on the planet employing the latest technological resources that allow remote consultations and accurate diagnoses. 3D printed medicines offer pharmacotherapeutic treatment as a response to a virtual prescription, paving the way for digital pharmacy [28], which completes the care cycle that can definitively mark the 21st century (Figure 1).

Despite the fast development of 3D printing in the pharmaceutical field and the market release of the first 3D printed drug product, Spritam^®^ (levetiracetam), there are technical and regulatory issues that need to be addressed [29]. This work aimed to update the theme of FDM 3D printing used for elaborate drug delivery devices, discussing the challenges and possible solutions that could allow this technology to enter the market.

## 2. The Versatility of FDM 3D Printing for Drug-Delivery Devices

FDM 3D printers can produce a wide range of different drug delivery devices, as evidenced by recently published scientific reports on the subject. A search of the combined terms, “3D printing”, “FDM” and “drug” in SciFinder^®^ for 2014 to 2018 resulted in 54 papers on the subject. These works give a glimpse of the technology’s potential (Figure 2).

As expected, the majority of studies explored the development of oral dosage forms, with tablets accounting for the largest share of the pie (63%), followed by capsules (11%). Oral pharmaceutical presentations represent more than 40% of the drug products in the market [30]. Simple control of printing variables can offer interesting therapeutic advantages to 3D printed drug products. Precision in dosage personalization is undoubtedly one of the great benefits of FDM 3D printing technology, as opportunely explored in the case of warfarin tablets. This active pharmaceutical ingredient was printed in tailored doses safely administered to rats, eliminating the need to split and facilitate the progression and regression of doses, as is usually employed in treatments with this drug [31].

Another distinguished approach was the printing of domperidone disks with low infill, increasing the drug time in the stomach through flotation, thus decreasing the frequency of tablet intake [32]. Moreover, several drugs can be easily associated with the same pharmaceutical unit, such as in the case of a “polypill” printed with intercalated layers of paracetamol and caffeine, leading to simultaneous release of both drugs [33]. FDM 3D printing could also produce immediate release tablets using distinct pharmaceutical grade polymers [34] or by adding gaps to the tablet design, called “gaplets”, which increase the porosity of the tablets [35].

Innovative geometries to increase patient compliance could also be created. Tests performed in vivo have revealed that patients are open to experimenting with new geometries, such as in donut form (torus) [6]. Pediatric dosage forms imitating candy may improve children’s acceptance of oral forms and the extrusion and printing processes using polymers may help mask the bitterness of active pharmaceutical ingredients [36]. Control of tablet shapes can also help modulate drug-release rates, which are dependent on surface area/volume of the tablet [7]. Figure 3 shows different shapes of FDM printed tablets and capsules.

In addition, 3D printing of capsules has advantageous performances. Studies have revealed that a capsule combining two different polymer compartments can produce a two-pulse release kinetic [37] and dual release for caffeine [38]. Hollow printed capsules containing complex compartments filled with liquid metformin, where the liquid formulation is not exposed to heating, lead to controlled drug release by capsule dissolution rate [39]. Printed capsules may also create different release rates of drug solutions by changing the shell thickness and core volume [40], whereas for printed cores, the infill and polymeric matrix even helps achieve a zero-order release rate [10]. Zero-order release was also reached in a three-part donut shaped tablet composed of polymeric water insoluble outside layers and a soluble polymeric drug loaded center [41].

Other oral forms printed using FDM are oral films and medicinal mouthguards, for example, aripiprazole oral films printed by 3D FDM that have improved drug dissolution rate using a porous polymeric matrix [42]; and the new personalized drug delivery device, shaped as a mouthguard, which contains clobetasol propionate to treat mouth inflammation [2].

Dermal adhesives for cutaneous drug delivery have also been produced using the same technique. Printed polylactic acid microneedles devices have been shown to be able to pierce porcine skin and deliver a model drug [43]. Vaginal rings and drug implants represented 2% and 7%, respectively, of studies on printed drug delivery devices (Figure 2). Printed progesterone vaginal rings in different shapes have distinct dissolution rates based on surface area/volume and release the drug over a period of one week [3]. A polylactic acid sub cutaneous implant for sustained release of disulfiram [44] and even ethylene vinyl acetate intrauterine devices loaded with indomethacin [45] have also been printed using 3D FDM.

Other complex delivery systems correspond to 6% of the devices described in Figure 2. For example, ‘tablet-in-devices’, which were developed to keep the riboflavin tablet floating in stomach acid for a longer period, enhances drug absorption and ensures a sustained release [46]. ‘Dual-compartmental dosage unit’ can combine two incompatible drugs (rifampicin and isoniazid) used in tuberculosis treatment, using polylactic acid to separate the drug filaments and generate distinct dissolution patterns [47].

3D printing technology has proven, therefore, to be capable of producing very complex anatomical shapes, with multiple active pharmaceutical ingredients and different release kinetics. However, despite this research, this topic is not exhaustive, and new pharmaceutical devices could be produced in future to deliver drugs to specific body requirements. Drug-loaded contact lens to treat eye disorders, pharmaceutical polymeric nails to treat fungal infections, and drug delivery head caps to treat baldness are obvious alternatives that have not been tested yet.

## 3. Adaptations of FDM 3D Printer for Pharmaceutical Production

As mentioned above, FDM printing is the most researched technique for 3D-printed drug delivery devices today, when compared to other 3D printing techniques such as selective laser sintering and powder bed. The equipment is affordable, easy to operate, and shows high print accuracy and reproducibility [22]. MakerBot^®^ (USA), Multirap M420^®^ (Germany) and Prusa i3^®^ (Czech Republic) are printer brands used in several studies; and their process variables, such as temperature, speed and infill, have been correlated with pharmaceutical production variables [22,48].

Despite this, no commercial model is available for pharmaceutical use. Moreover, recent studies point out that numerous adaptations of commercial machines are needed to meet pharmaceutical production requirements [49]. Figure 4 shows the main FDM printer parts that require adjustments.

The spool containing the extruded drug loaded filament (Figure 4a) is attached to the printer through a tube, from where it reaches the equipment nozzle through a gear system [50]. In several printers, the filament coil in the spool is not protected from particles or humidity during the printing process. This exposure could lead to cross-contamination of filament spools, which could ideally be reused. To solve this problem, a closed compartment could be connected to the printer, covering the spool attachment. Store boxes for filaments are now available; they can be coupled with the 3D printer to protect the filaments from moisture, dust contamination; and keep them heated for a better printing result (eBox^®^, eSUN, China). The printer enclosure (Figure 4b) should also be sealed against contaminants, as in MakerBot Replicator 2x^®^ (USA), protecting the printed drug delivery device and eliminating the need for a laminar airflow or strict particle control of the production area [38].

In order to meet Good Manufacturing Practices, all the printer parts, such as extruder head, nozzle and build platform (Figure 4c–e), which are in direct contact with the drug-loaded filament should be made of an inert material that can be easily cleaned—accordingly, stainless steel is the most recommended. Similar to what happens in hot-melt extruders, the use of cleaning polymers may be more efficient in removing residues from the machine than the use of solvent and chemical products, such as water and soap. In addition, the mechanical parts of the FDM printer, such as the motors (Figure 4f), need to be completely closed, preventing lubricant oil from spilling over the product [38].

The thermal processing of the formulation involved in FDM 3D printing and its risks to drug stability cannot be ignored. In fact, stability issues for thermosensitive drugs have already been noted by initial application of this technology [51]. The problem has been addressed by using polymers with low glass transition temperature or by using plasticizers, which can reduce the polymers glass transition and consequently the nozzle temperature [34,52]. Taking this into consideration, an important point for pharmaceutical use concerns the need of a more precise and sensitive control of the heater’s temperature (Figure 4g). Overheating could lead to modifications of the polymer viscoelasticity compromising drug control release and, eventually, stability of the drug.

Another operational problem with some FDM 3D printers is the lack of flexibility in the size of the nozzle (Figure 4d), since commercial filaments have a standard 1.75 mm diameter. However, for pharmaceutical use, a wide range of polymeric materials is extruded generating diameter oscillations due to their viscoelastic characteristics. In that case, it is necessary to choose a printer with adjustable nozzles [53]. Furthermore, an optimized FDM printer with multiple nozzles could improve the printing time of a batch or even produce devices with multiple APIs without the need to change filaments during the process [33]. Currently there are certain commercial models that comply with those needs, such as Stacker S4^®^ (Stacker Corp., USA), an industrial multi nozzle 3D printer capable of printing four objects at the same time, and RoVa3D^®^ (ORD Solutions Inc., Canada), which possesses 0.2 mm, 0.35 mm, 0.5 mm, 0.7 mm, 1.0 mm diameter nozzles and can print up to five filaments simultaneously.

In order to extend the possibilities of FDM 3D printing and to increase the automation of the process, it would be desirable to fill scaffolds printed with a liquid or semisolid containing the drug using a piston or syringe coupled to the printer [40,54]. For instance, the use of thermogelling materials such as poloxamers may be of particular utility in modulating drug release [55]. The HYREL 3D company (USA) developed an interesting solution for the same—a set of different printer heads compatible with their machines. The modular heads are assembled to the printers to allow the introduction of different materials in the 3D device, such as hot flow heads for FDM filaments, cold and warm flow heads for pastes and resins, and cold flow syringe heads for liquids and gels.

The software used to control the printer (Figure 4h) should be designed to receive electronic prescriptions from the physician’s office and to suggest the most recommended conditions of printing. The more complex information, such as nozzle and platform temperature, speed, layer height and infill, would be supplied by pharmacy technicians, based on filament manufacturer information. After training, the professional should be qualified to operate the machine. The 3D design of devices would be preferably pre-selected from a database, thus saving time when defining device shape [56,57].

## 4. Integrated Production Process

Diversified drug delivery devices using 3D FDM technology are being developed at a fast pace by dozens of researches groups in different parts of the globe [58]. Nevertheless, another step has to be taken to allow the commercial viability of this technology. In the light of previous works and considering the extrusion process already used by the pharmaceutical industry, a partnership between pharmaceutical industries and compounding pharmacies in a complementary production chain appears as the most viable alternative to create a new pathway to the market (Figure 5).

### 4.1. Pharmaceutical Industry: Filament Production by Hot-Melt Extrusion

The production of drug-loaded filaments used to feed FDM 3D printers consists of a known process with industrial production profiles. The routine of filament production follows three major steps (Figure 5): First, the components of the batch are mixed, for example in an industrial V-blender. Next, the hot-melt extruder, fed with the component’s mixture, produces the drug-loaded filaments by shear and heating. Finally, filament bulk is packed in smaller spools. This intermediate pharmaceutical product should be hermetically packed to prevent product deterioration until it reaches compounding pharmacies.

In the initial phase, the industry dedicated to producing the filaments should conduct research and development for each product, based on the selected drug and desired drug release profile. In this stage, formulation composition, as well as extrusion process, should be defined by following quality-by-design planning [59].

Hot-melt extrusion is an already stablished industrial production process for drugs, but the finer nuances of fabrication of printable filaments needs to be studied further. Thermal and rheological studies should be performed to determine the compatibility between the components and their suitability for the extrusion processing and for the 3D FDM printing. Then, stability studies should be conducted to determine product shelf-life. Different modified drug release profiles can be achieved with the manufacturing of filaments from polymers with delimited solubilization characteristics (fast, slow, pH dependent, etc.).

Filament diameter is crucial for the printing process; improper size can cause the extrudate to clog or lead to a lower feeding rate [60]. Some hygroscopic polymers can cause diameter enlargement, compromising passage of the filament through the printer mechanism [41]. In addition, the heating process could cause diameter deformities, which is why an external pulley with a cooling system should be attached to the extruder dye end in some cases [53].

Routine quality control tests may include organoleptic characteristics, dimensions, rheological properties, tensile strength, thermal behavior, and drug content. In addition, drug release should be evaluated using dissolution apparatus or Franz diffusion cells [61,62].

### 4.2. Digital Pharmacy: 3D FDM Printing of Personalized Drug Products

Current compounding pharmacies have the necessary infrastructure to produce 3D printed drug delivery devices. In fact, their usual layout is appropriate for FDM pharmaceutical printer installation without requiring major adaptations or masonry work [30]. In a compounding pharmacy, the equipment could be set over existing benches in the solid and liquid preparation rooms, requiring only a power source to operate [63].

The printers could be networked by one or more computers equipped with the necessary interfaces in a central control room. The prescriptions arrive from the doctor’s offices remotely, and after authorization of the pharmacy administration and revision by the pharmacist they are sent for impression to one of the available printers. Thus, with investment in the purchase of printers, software, and personnel training, it is possible to transform a regular compounding pharmacy into a digital one. Figure 6 shows the layout of a hypothetical compounding pharmacy with FDM 3D printers.

The drug delivery production process should depend on three major steps (Figure 5). First, trained technicians with access to the compounding area should set printer parameters, such as infill, velocity, resolution, temperature, and others, using the pharmaceutical printer software [64]. The information provided by the filament manufacturer and the prescription should serve as guidance. In addition, an adequate design (shape) for the drug delivery device needs to be selected from the database [7]. It is important that the FDM printer is compatible with the filament manufacturer’s specifications, so that the intermediate product can be safely used and validated to guarantee process specifications [49].

The second step should include printer preparation and spool attachment. To avoid wastage and exposure of the entire spool to the printing process, the needed amount of filament for the desired batch should be cut and transferred to a smaller spool. The rest of the filament should be hermetically restored using a vacuum device. A FDM-adapted pharmaceutical printer with the modifications discussed before should produce the desired drug tailored for a specific patient. After printing, the product batch should be removed, and the technician could proceed to step three. In this stage, the produced drug delivery device is packed and labeled according to current legislation and dispensed to the patient in the reception room [65]. Before the printer is used again with a different filament, it must be cleaned following a validated method [66].

Besides the use of polymers with characteristics that modulate the drug release from the printed medicine, drug release kinetics can also be directed by adjusting printing parameters such as infill percentage, infill pattern or print speed. In the case of drug combinations in the same dosage form, it is possible to alternate the filaments that are feeding the printer in order to build the desired structure. An innovative approach to do this can be performed with the aid of the Palette 2 device (Mosaic Manufacturing Ltd., Canada), in which different filaments are combined into one that will feed the 3D printer. This filament produced from the merger of numerous fragments of several filaments has its composition defined according to the 3D structure to be built in the printer.

The quality control required for the finished drug products should be the same as those currently required for common solid preparations in compounding pharmacies, which includes average weight and organoleptic characteristics. However, due to the high automation of the 3D printing process and the small number of production steps, there is a lower risk of human error, and consequently a noticeable gain in the safety of printed drug products. Extra tests to determine drug delivery device characteristics, such as hardness, friability, drug content, and drug release could be applied to pilot batches or by sampling in accordance with each country’s regulatory demands [37,38,67].

Several studies have shown innumerable cases of intoxication related to the use of medicines produced in compounding pharmacies, resulting, for example, from errors in weighing [18]. This scenario may be overcome by the introduction of 3D printing of drug products. Moreover, new analytical alternatives have been probed for quality control of the 3D printed structures. One is the use of tools with viability for batch-to-batch analysis; such as near infrared spectroscopy, which can perform drug content determinations with a sensitivity comparable to that of chromatographic methods [1]; and terahertz pulsed imaging, which allows acquisition of single-depth scans in a few milliseconds, providing information on the microstructure of the printed devices [54]. These new analytical approaches can make a significant contribution to the safety of printed pharmaceutical preparations.

## 5. Patent and Regulatory Limitations

In the few last years, 3D printing of medical devices has gained worldwide attention; in particular, products such as cranial implants, artificial knees, and spine prosthesis, which are personalized for each patient. Such products are marketed under current FDA regulations following their similarities with already existing medical devices [68]. In 2017, the FDA released guidelines for the manufacturing of medical devices and implants; however, there are currently no regulatory guidelines on the 3D printing of other products [69,70].

In 2015, the FDA released the first 3D printed drug product, Spritam^®^ (levetiracetam) [71]. This great technological step led to the increase of research on 3D printing technology to produce drug delivery devices. However, despite the fast development in the field, there are legal and regulatory issues that need to be addressed [72].

Spritam^®^ is an oral, fast disintegrating tablet, approved by existing legislation for largescale industrial production. The 3D printing process improved upon a disintegrating process, given that the drug contained a known active pharmaceutical ingredient (levetiracetam), in a permitted dosage (up to 1000 mg) to treat a stablished condition (epilepsy) [20,73]. 3D printing in an industrial scale presents benefits, as the design of complex geometries, when compared to other technologies, such as tableting, is not as competitive. Tailored formulations, ‘polypills’ and orphan medications produced in small batches can reach places the pharmaceutical industry cannot envision [17].

Many 3D printing technologies have lost their patents over the last decade, which was a decisive factor in making these machines more accessible to the public and to the pharmaceutical industry [56]. The patentability process, especially with regard to intellectual property rights involving 3D printed drug products, should be granted to innovative processes or products. The patent owner has exclusivity on the product or process until the concession expires; in the meantime, other manufactures may not produce, use, or sell without the owner’s authorization [74].

Despite this patent right, extemporaneous formulations produced at compounding pharmacies prescribed by professionals to a specific patient are exempted and do not configure patent violation, according to the intellectual property law of several countries, such as UK and Brazil [74]. Unlike in the US and Europe, where compounding pharmacies represent a small share of the market, in Brazil there are about 16,000 compounding pharmacies, which handle more than 60 million prescriptions per year [75]. If the market for compounding pharmacies is not a threat to large pharmaceutical corporations, this technological leap by digital pharmacies can change the global market scenario. However, this could provoke major legal disputes. Other drug delivery technologies already patented and vastly researched on could be rapidly brought back, not because of any drawback of the new technology, but for economic reasons, e.g., microneedles, active drug delivery patches using iontophoresis or sonophoresis [76]. The pharmaceutical industry is known to manage risk by broadening the use of old technologies, instead of accepting novel products. However, the difference here is in the demand chain—not from the industry, which carries the burden of educating and promoting the product; but by patients or healthcare personnel who place demands on small compounding pharmacies; and ultimately on the industry for supply of raw material. Hence, with growing research and small investments by compounding pharmacy owners, the industry might be forced to respond to such demands. The risk then would be consequently diminished, showcasing optimistic prognostics.

## 6. Conclusions

FDM 3D printing is a versatile technology widely studied for the production of multiple drug delivery devices. Research groups over the world are currently working to identify the nuances of the production process, and despite the great progress made so far, palpable planning to bring these products to life is necessary. The potential of 3D printing for the development of personalized drug products is undeniable; however, machine adaptations are fundamental for proper pharmaceutical use. In addition, a viable production process needs the co-participation of the pharmaceutical industry (to extrude filaments on a large scale), and digital pharmacies (to print drugs according to patient-specific prescriptions). Finally, regulatory and patent agencies should work together with companies to carve a solid path into the market.

## Figures and Tables

**Figure 1 pharmaceutics-11-00128-f001:**
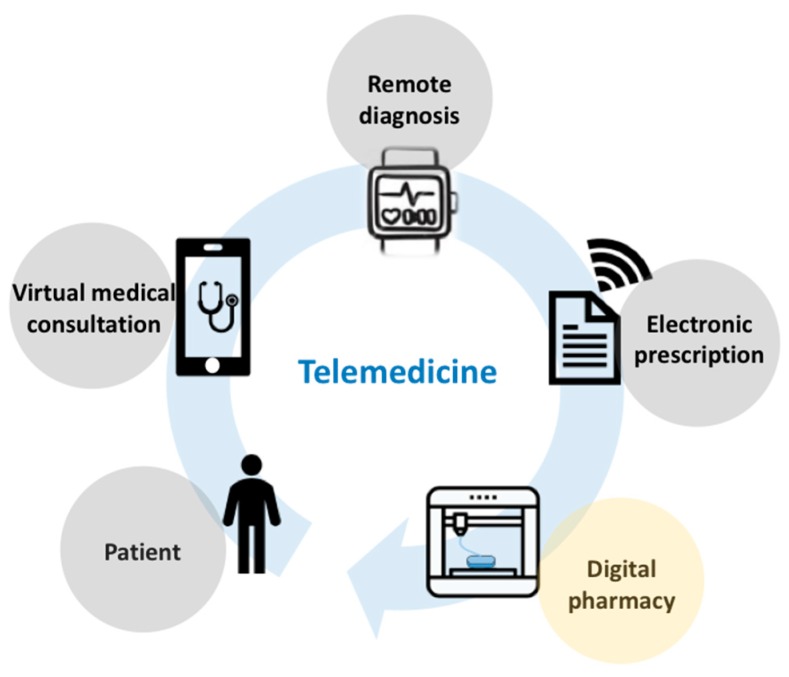
Telemedicine care cycle with insertion of digital pharmacy.

**Figure 2 pharmaceutics-11-00128-f002:**
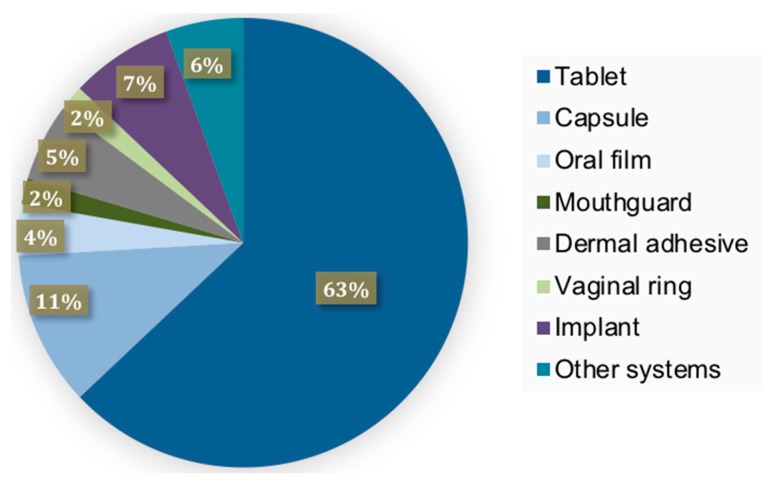
The share of drug-delivery devices (*n* = 54) that could be printed using fused deposition modeling 3D printing, as published in research papers between years 2014 and 2018 (SciFinder^®^).

**Figure 3 pharmaceutics-11-00128-f003:**
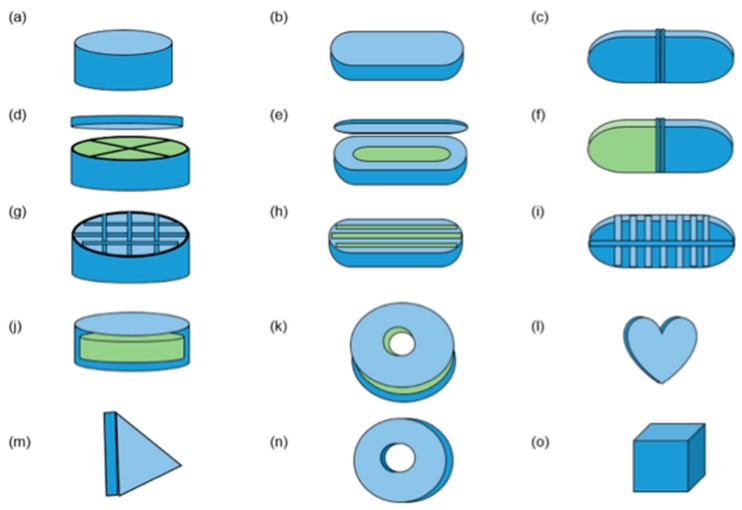
Different shapes of oral drug-delivery devices produced using fused deposition modeling 3D printing (blue = API 1, green = API 2). (**a**) [32], (**b**) [31], (**c**) [38], (**d**) [39], (**e**) [40], (**f**) [37], (**g**) [34], (**h**) [33], (**i**) [35], (**j**) [10], (**k**) [41], (**l**) [36], (**m**) [33], (**n**) [6], (**o**) [6].

**Figure 4 pharmaceutics-11-00128-f004:**
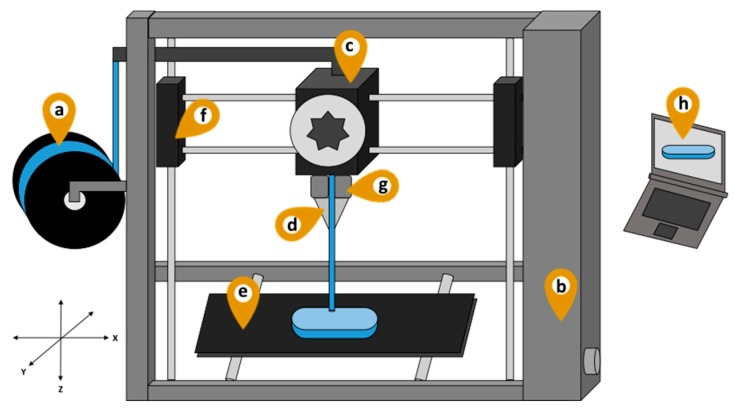
Schematic representation of a pharmaceutical fused deposition modeling 3D printer with indications of the specific points in which adaptations will be required for pharmaceutical production. (a) Spool, (b) printer enclosure, (c) extruder head, (d) nozzle, (e) build platform, (f) motor, (g) heater, (h) 3D design software.

**Figure 5 pharmaceutics-11-00128-f005:**
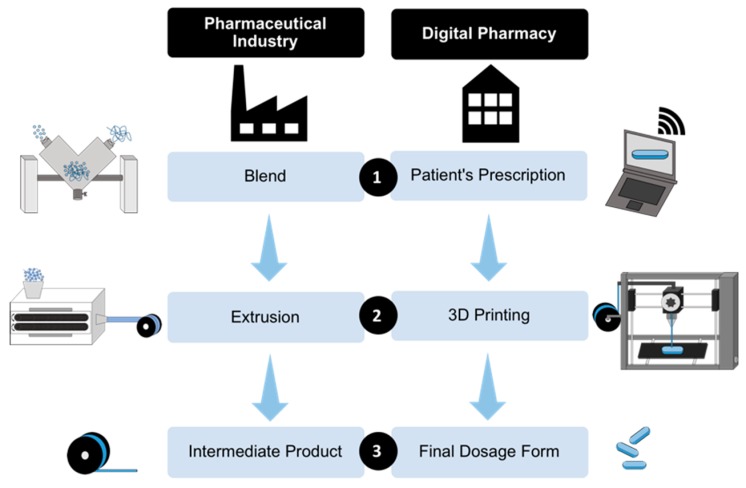
Schematic steps required for the industrial production of filaments and the elaboration of personalized drug delivery device in digital pharmacies.

**Figure 6 pharmaceutics-11-00128-f006:**
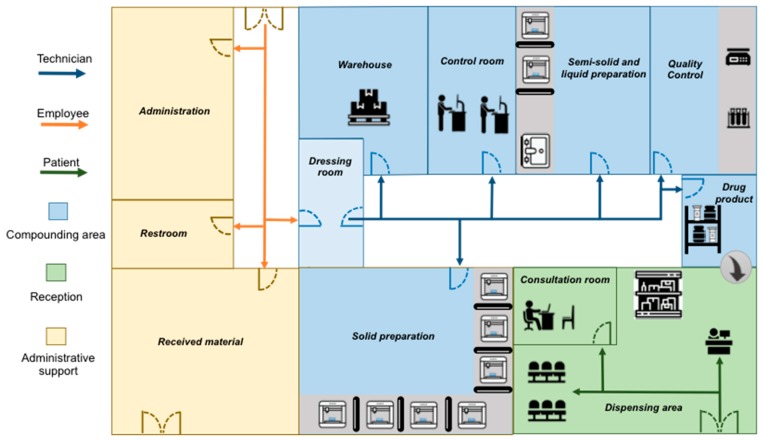
Digital pharmacy layout equipped with FDM 3D printers.

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
