# Peer review of "The Digital Pharmacies Era: How 3D Printing Technology Using Fused Deposition Modeling Can Become a Reality"

_pharmaceutics, 2019, doi:10.3390/pharmaceutics11030128_

Reviewer 1 Report

The manuscript titled “The digital pharmacies era: how 3D printing technology using fused deposition modeling can become a reality” discusses the state of art of the FDM 3D printing from a pharmaceutical prospective. Generally speaking, the review is well-written. However, it seems that the authors are not well updated with the field as many of the suggestions in sections 3 and 4 have already been implemented in multiple commercial FDM 3D printers but this is not mentioned or discussed really. Moreover, they draw a positive image of the FDM technology, leaving aside important discussions such as problems associated with elevated temperatures and its effect on drug stability. In addition, they talk about FDM being the most used technology but they never really explain the reason behind its prevalent use compared to other technologies.

More detailed comments include:

Line 22: on “one” hand

Line 38: production parks?

Line 47: I believe this paragraph could use some extra “specific” referencing instead referencing a review article, e.g.:

Multiple actives:

Okwuosa, T.C.; Pereira, B.C.; Arafat, B.; Cieszynska, M.; Isreb, A.; Alhnan, M.A. Fabricating a Shell-Core Delayed Release Tablet Using Dual FDM 3D Printing for Patient-Centred Therapy. Pharm Res 201734, 427-437, doi:10.1007/s11095-016-2073-3.

Gioumouxouzis, C.I.; Baklavaridis, A.; Katsamenis, O.L.; Markopoulou, C.K.; Bouropoulos, N.; Tzetzis, D.; Fatouros, D.G. A 3D printed bilayer oral solid dosage form combining metformin for prolonged and glimepiride for immediate drug delivery. Eur. J. Pharm. Sci. 2018120, 40-52, doi:10.1016/j.ejps.2018.04.020.

Flexible dosages:

Liang, K.; Carmone, S.; Brambilla, D.; Leroux, J.C. 3D printing of a wearable personalized oral delivery device: A first-in-human study. Science Advances 2018, 4, eaat2544, doi:10.1126/sciadv.aat2544.

Fu, J.; Yin, H.; Yu, X.; Xie, C.; Jiang, H.; Jin, Y.; Sheng, F. Combination of 3D printing technologies and compressed tablets for preparation of riboflavin floating tablet-in-device (TiD) systems. International Journal of Pharmaceutics 2018, 549, 370-379, doi:10.1016/j.ijpharm.2018.08.011.

Muwaffak, Z.; Goyanes, A.; Clark, V.; Basit, A.W.; Hilton, S.T.; Gaisford, S. Patient-specific 3D scanned and 3D printed antimicrobial polycaprolactone wound dressings. Int J Pharm 2017527, 161-170, doi:10.1016/j.ijpharm.2017.04.077.

Goyanes, A.; Det-Amornrat, U.; Wang, J.; Basit, A.W.; Gaisford, S. 3D scanning and 3D printing as innovative technologies for fabricating personalized topical drug delivery systems. J. Control. Release 2016,234, 41-48, doi:10.1016/j.jconrel.2016.05.034.

Shapes:

Martinez, P.R.; Goyanes, A.; Basit, A.W.; Gaisford, S. Influence of Geometry on the Drug Release Profiles of Stereolithographic (SLA) 3D-Printed Tablets. AAPS PharmSciTech. 201819, 3355-3361, doi:10.1208/s12249-018-1075-3.

Goyanes, A.; Martinez, P.R.; Buanz, A.; Basit, A.; Gaisford, S. Effect of geometry on drug release from 3D printed tablets. Int J Pharm 2015494, 657-663, doi:10.1016/j.ijpharm.2015.04.069.

Fu, J.; Yu, X.; Jin, Y. 3D printing of vaginal rings with personalized shapes for controlled release of progesterone. International Journal of Pharmaceutics 2018539, 75-82, doi:10.1016/j.ijpharm.2018.01.036.

Goyanes, A.; Scarpa, M.; Kamlow, M.; Gaisford, S.; Basit, A.W.; Orlu, M. Patient acceptability of 3D printed medicines. International Journal of Pharmaceutics 2017530, 71-78, doi:10.1016/j.ijpharm.2017.07.064.

Modulated release kinetics:

Goyanes, A.; Fina, F.; Martorana, A.; Sedough, D.; Gaisford, S.; Basit, A.W. Development of modified release 3D printed tablets (printlets) with pharmaceutical excipients using additive manufacturing. International Journal of Pharmaceutics 2017527, 21-30, doi:10.1016/j.ijpharm.2017.05.021.

Tagami, T.; Nagata, N.; Hayashi, N.; Ogawa, E.; Fukushige, K.; Sakai, N.; Ozeki, T. Defined drug release from 3D-printed composite tablets consisting of drug-loaded polyvinylalcohol and a water-soluble or water-insoluble polymer filler. International Journal of Pharmaceutics 2018543, 361-367, doi:10.1016/j.ijpharm.2018.03.057.

Zhang, J.; Feng, X.; Patil, H.; Tiwari, R.V.; Repka, M.A. Coupling 3D printing with hot-melt extrusion to produce controlled-release tablets. Int J Pharm 2017519, 186-197, doi:10.1016/j.ijpharm.2016.12.049.

Goyanes, A.; Buanz, A.B.; Basit, A.W.; Gaisford, S. Fused-filament 3D printing (3DP) for fabrication of tablets. International Journal of Pharmaceutics 2014476, 88-92, doi:10.1016/j.ijpharm.2014.09.044.

Bloomquist, C.J.; Mecham, M.B.; Paradzinsky, M.D.; Janusziewicz, R.; Warner, S.B.; Luft, J.C.; Mecham, S.J.; Wang, A.Z.; DeSimone, J.M. Controlling release from 3D printed medical devices using CLIP and drug-loaded liquid resins. Journal of Controlled Release 2018278, 9-23, doi:10.1016/j.jconrel.2018.03.026.

Line 50-53: I believe that paragraph benefit from referencing the following paper:

Awad, A.; Trenfield, S.J.; Goyanes, A.; Gaisford, S.; Basit, A.W. Reshaping drug development using 3D printing. Drug Discovery Today 2018,23, 1547-1555, doi:10.1016/j.drudis.2018.05.025.

Line 58-59: it is dangerous to say 3d printing is “safer” as this has not been fully established yet. I believe personalisation of medication is safe. Also, I believe there are more suitable papers that should be referenced in this context:

Mukhopadhyay, S.; Poojary, R. A review on 3D printing: Advancement in healthcare technology. In Proceedings of 2018 Advances in Science and Engineering Technology International Conferences (ASET), Feb. 6 2018-April 5 2018; pp. 1-5.

Trenfield, S.J.; Awad, A.; Goyanes, A.; Gaisford, S.; Basit, A.W. 3D Printing Pharmaceuticals: Drug Development to Frontline Care. Trends Pharmacol. Sci. 201839, 440-451, doi:10.1016/j.tips.2018.02.006.

Norman, J.; Madurawe, R.D.; Moore, C.M.V.; Khan, M.A.; Khairuzzaman, A. A new chapter in pharmaceutical manufacturing: 3D-printed drug products. Advanced Drug Delivery Reviews 2017108, 39-50, doi:https://doi.org/10.1016/j.addr.2016.03.001.

Line 105: “frequency” of tablet intake.

Line 131,134 and figure 2: “mouthguards” not mouthgards.

Line 169: “require” or “could use” or “could benefit from” adjustments not suffer

Line 172: there are now temperature-controlled filament storage boxes that can be used that will keep the filaments in a suitable condition even while printing. 

Lines 199-201: again, there are now printer’s with multiple heads, e.g. Stacker S4 has 4 heads that can be used for multi-material printing. The authors should give true examples of how this platform is developing, rather than repeating the same old stuff.

Line 202-204: There now even printer that can combine FDM printing with semisolid extrusion or liquid-based syringes, e.g. Hyrel printer.

Section 3: what about issues with drug degradation due to elevated temperatures? This has been found to be an issue with FDM:

Goyanes, A.; Buanz, A.B.; Hatton, G.B.; Gaisford, S.; Basit, A.W. 3D printing of modified-release aminosalicylate (4-ASA and 5-ASA) tablets. Eur J Pharm Biopharm 201589, 157-162, doi:10.1016/j.ejpb.2014.12.003.

Yes, this has been resolved in the following papers, e.g.:

Kempin, W.; Domsta, V.; Grathoff, G.; Brecht, I.; Semmling, B.; Tillmann, S.; Weitschies, W.; Seidlitz, A. Immediate Release 3D-Printed Tablets Produced Via Fused Deposition Modeling of a Thermo-Sensitive Drug. Pharmaceutical Research 201835, 124, doi:10.1007/s11095-018-2405-6.

Kollamaram, G.; Croker, D.M.; Walker, G.M.; Goyanes, A.; Basit, A.W.; Gaisford, S. Low temperature fused deposition modeling (FDM) 3D printing of thermolabile drugs. International Journal of Pharmaceutics 2018545, 144-152, doi:https://doi.org/10.1016/j.ijpharm.2018.04.055.

Yet, it still should be discussed.

Section 4: I do not believe preparing filaments using HME is “simple”. The process is rather complex, especially with pharmaceutical excipients, as it is hard to get filaments with characteristics suitable for FDM printing. Also, if you are storing filaments have many possible combinations of drugs+polymers can you store? If you are personalising medications the drugs would differ, so will the polymer it is dispersed in...personalising medications is not just about personalizing the dose, as it also involves personalizing the drug release, which can be controlled but selecting a suitable polymer. So, how many possible combinations can be stored? What about stability issues? Also, storage space? 

Line 244-246: what about simple tests of batch-to-batch quality control please discuss further.

Author Response

The manuscript titled “The digital pharmacies era: how 3D printing technology using fused deposition modeling can become a reality” discusses the state of art of the FDM 3D printing from a pharmaceutical prospective. Generally speaking, the review is well-written. However, it seems that the authors are not well updated with the field as many of the suggestions in sections 3 and 4 have already been implemented in multiple commercial FDM 3D printers but this is not mentioned or discussed really. Moreover, they draw a positive image of the FDM technology, leaving aside important discussions such as problems associated with elevated temperatures and its effect on drug stability. In addition, they talk about FDM being the most used technology but they never really explain the reason behind its prevalent use compared to other technologies.

Response: We appreciated your careful evaluation of our manuscript. The responses to your comments are listed and addressed one-by-one below.

Minor Points

1. Line 22: on “one” hand

Response: The correction was performed. 

2. Line 38: production parks?

Response: Thank you for the correction. We replaced by the term “industrial facilities”.

3. Line 47: I believe this paragraph could use some extra “specific” referencing instead referencing a review article, e.g.: 

Multiple actives:

Okwuosa, T.C.; Pereira, B.C.; Arafat, B.; Cieszynska, M.; Isreb, A.; Alhnan, M.A. Fabricating a Shell-Core Delayed Release Tablet Using Dual FDM 3D Printing for Patient-Centred Therapy. Pharm Res 2017, 34, 427-437, doi:10.1007/s11095-016-2073-3.

Gioumouxouzis, C.I.; Baklavaridis, A.; Katsamenis, O.L.; Markopoulou, C.K.; Bouropoulos, N.; Tzetzis, D.; Fatouros, D.G. A 3D printed bilayer oral solid dosage form combining metformin for prolonged and glimepiride for immediate drug delivery. Eur. J. Pharm. Sci. 2018, 120, 40-52, doi:10.1016/j.ejps.2018.04.020.

Flexible dosages:

Liang, K.; Carmone, S.; Brambilla, D.; Leroux, J.C. 3D printing of a wearable personalized oral delivery device: A first-in-human study. Science Advances 2018, 4, eaat2544, doi:10.1126/sciadv.aat2544.

Fu, J.; Yin, H.; Yu, X.; Xie, C.; Jiang, H.; Jin, Y.; Sheng, F. Combination of 3D printing technologies and compressed tablets for preparation of riboflavin floating tablet-in-device (TiD) systems. International Journal of Pharmaceutics 2018, 549, 370-379, doi:10.1016/j.ijpharm.2018.08.011.

Muwaffak, Z.; Goyanes, A.; Clark, V.; Basit, A.W.; Hilton, S.T.; Gaisford, S. Patient-specific 3D scanned and 3D printed antimicrobial polycaprolactone wound dressings. Int J Pharm 2017, 527, 161-170, doi:10.1016/j.ijpharm.2017.04.077.

Goyanes, A.; Det-Amornrat, U.; Wang, J.; Basit, A.W.; Gaisford, S. 3D scanning and 3D printing as innovative technologies for fabricating personalized topical drug delivery systems. J. Control. Release 2016,234, 41-48, doi:10.1016/j.jconrel.2016.05.034.

Shapes:

Martinez, P.R.; Goyanes, A.; Basit, A.W.; Gaisford, S. Influence of Geometry on the Drug Release Profiles of Stereolithographic (SLA) 3D-Printed Tablets. AAPS PharmSciTech. 2018, 19, 3355-3361, doi:10.1208/s12249-018-1075-3.

Goyanes, A.; Martinez, P.R.; Buanz, A.; Basit, A.; Gaisford, S. Effect of geometry on drug release from 3D printed tablets. Int J Pharm 2015, 494, 657-663, doi:10.1016/j.ijpharm.2015.04.069.

Fu, J.; Yu, X.; Jin, Y. 3D printing of vaginal rings with personalized shapes for controlled release of progesterone. International Journal of Pharmaceutics 2018, 539, 75-82, doi:10.1016/j.ijpharm.2018.01.036.

Goyanes, A.; Scarpa, M.; Kamlow, M.; Gaisford, S.; Basit, A.W.; Orlu, M. Patient acceptability of 3D printed medicines. International Journal of Pharmaceutics 2017, 530, 71-78, doi:10.1016/j.ijpharm.2017.07.064.

Modulated release kinetics:

Goyanes, A.; Fina, F.; Martorana, A.; Sedough, D.; Gaisford, S.; Basit, A.W. Development of modified release 3D printed tablets (printlets) with pharmaceutical excipients using additive manufacturing. International Journal of Pharmaceutics 2017, 527, 21-30, doi:10.1016/j.ijpharm.2017.05.021.

Tagami, T.; Nagata, N.; Hayashi, N.; Ogawa, E.; Fukushige, K.; Sakai, N.; Ozeki, T. Defined drug release from 3D-printed composite tablets consisting of drug-loaded polyvinylalcohol and a water-soluble or water-insoluble polymer filler. International Journal of Pharmaceutics 2018, 543, 361-367, doi:10.1016/j.ijpharm.2018.03.057.

Zhang, J.; Feng, X.; Patil, H.; Tiwari, R.V.; Repka, M.A. Coupling 3D printing with hot-melt extrusion to produce controlled-release tablets. Int J Pharm 2017, 519, 186-197, doi:10.1016/j.ijpharm.2016.12.049.

Goyanes, A.; Buanz, A.B.; Basit, A.W.; Gaisford, S. Fused-filament 3D printing (3DP) for fabrication of tablets. International Journal of Pharmaceutics 2014, 476, 88-92, doi:10.1016/j.ijpharm.2014.09.044.

Bloomquist, C.J.; Mecham, M.B.; Paradzinsky, M.D.; Janusziewicz, R.; Warner, S.B.; Luft, J.C.; Mecham, S.J.; Wang, A.Z.; DeSimone, J.M. Controlling release from 3D printed medical devices using CLIP and drug-loaded liquid resins. Journal of Controlled Release 2018, 278, 9-23, doi:10.1016/j.jconrel.2018.03.026.

Response: We now used the recommended references as suggested.

4. Line 50-53: I believe that paragraph benefit from referencing the following paper:

Awad, A.; Trenfield, S.J.; Goyanes, A.; Gaisford, S.; Basit, A.W. Reshaping drug development using 3D printing. Drug Discovery Today 2018,23, 1547-1555, doi:10.1016/j.drudis.2018.05.025.

Response: We added the reference as suggested.

5. Line 58-59: it is dangerous to say 3d printing is “safer” as this has not been fully established yet. I believe personalisation of medication is safe. Also, I believe there are more suitable papers that should be referenced in this context:

Mukhopadhyay, S.; Poojary, R. A review on 3D printing: Advancement in healthcare technology. In Proceedings of 2018 Advances in Science and Engineering Technology International Conferences (ASET), Feb. 6 2018-April 5 2018; pp. 1-5.

Trenfield, S.J.; Awad, A.; Goyanes, A.; Gaisford, S.; Basit, A.W. 3D Printing Pharmaceuticals: Drug Development to Frontline Care. Trends Pharmacol. Sci. 2018, 39, 440-451,

Norman, J.; Madurawe, R.D.; Moore, C.M.V.; Khan, M.A.; Khairuzzaman, A. A new chapter in pharmaceutical manufacturing: 3D-printed drug products. Advanced Drug Delivery Reviews 2017, 108, 39-50, doi:https://doi.org/10.1016/j.addr.2016.03.001.

Response: We rebuilt this sentence and added more suitable references as recommended. Please see the revised manuscript on page 2 (lines 64-67).

“…The automation of the 3D printing process and particularly the high accuracy achieved by FDM technology make the printing of drug products potentially safer…”

6. Line 105: “frequency” of tablet intake.

Response: The word was corrected.

7. Line 131,134 and figure 2: “mouthguards” not mouthgards.

Response: The corrections were performed.

8. Line 169: “require” or “could use” or “could benefit from” adjustments not suffer

Response: We now used the term “require” in the sentence.

9. Line 172: there are now temperature-controlled filament storage boxes that can be used that will keep the filaments in a suitable condition even while printing. 

Response: Thank you. The information was updated and eBox®was used as an example. Please, see the revised manuscript on page 5.

“...This exposure could lead to a cross contamination of filament spools that would be reused in a next printing process. To solve this inconvenient, a closed compartment could be connected to the printer covering the spool attachment. In this sense, store boxes for the filaments are now available. Such store boxes can be coupled to the 3D printer, protecting the filaments from moisture, dust contamination and keeping them heated for a better printing result (eBox®, eSUN, China).”

10. Lines 199-201: again, there are now printer’s with multiple heads, e.g. Stacker S4 has 4 heads that can be used for multi-material printing. The authors should give true examples of how this platform is developing, rather than repeating the same old stuff.

Response: The information was updated. Stacker S4®and RoVa3D® printers were used as examples. Please, see the revised manuscript on page 6.

“…Currently there are certain commercial models that comply with those needs, such as Stacker S4®(Stacker Corp., USA), an industrial multi nozzle 3D printer capable of printing four objects at the same time and RoVa3D®(ORD Solutions Inc., Canada), which possesses 0.2, 0.35, 0.5, 0.7, 1.0 mm diameter nozzles and can print up to five filaments simultaneously.”

11. Line 202-204: There now even printer that can combine FDM printing with semisolid extrusion or liquid-based syringes, e.g. Hyrel printer.

Response: Thank you for the information. HYREL printer was used as an example. Please see page 7 of the revised manuscript. 

“The HYREL 3D company (USA) developed an interesting solution for that matter, a set of different printer heads compatible with their machines. The modular heads are assembled to the printers to allow the introduction of different materials in the 3D device, such as the hot flow heads for FDM filaments, the cold and warm flow heads for pastes and resins and the cold flow syringe heads for liquids and gels.”

12. Section 3: what about issues with drug degradation due to elevated temperatures? This has been found to be an issue with FDM:

Goyanes, A.; Buanz, A.B.; Hatton, G.B.; Gaisford, S.; Basit, A.W. 3D printing of modified-release aminosalicylate (4-ASA and 5-ASA) tablets. Eur J Pharm Biopharm 2015, 89, 157-162, doi:10.1016/j.ejpb.2014.12.003.

Yes, this has been resolved in the following papers, e.g.:

Kempin, W.; Domsta, V.; Grathoff, G.; Brecht, I.; Semmling, B.; Tillmann, S.; Weitschies, W.; Seidlitz, A. Immediate Release 3D-Printed Tablets Produced Via Fused Deposition Modeling of a Thermo-Sensitive Drug. Pharmaceutical Research 2018, 35, 124, doi:10.1007/s11095-018-2405-6.

Kollamaram, G.; Croker, D.M.; Walker, G.M.; Goyanes, A.; Basit, A.W.; Gaisford, S. Low temperature fused deposition modeling (FDM) 3D printing of thermolabile drugs. International Journal of Pharmaceutics 2018, 545, 144-152, doi:https://doi.org/10.1016/j.ijpharm.2018.04.055.

Yet, it still should be discussed.

Response: Thank you for the pertinent indication of the references. A complementary paragraph about these issues was added as well as the references. Please, see the revised manuscript on page 6.

“The thermal processing of the formulation involved in the FDM 3D printing and its risks to the drug stability cannot be ignored. In fact, stability issues for thermosensitive drugs have already been warned by the initial works using this technology [51]. The problem has been addressed by using polymers with low glass transition temperature or by using plasticizers, which can reduce the polymers glass transition and consequently the nozzle temperature [34,52].”

13. Section 4: I do not believe preparing filaments using HME is “simple”. The process is rather complex, especially with pharmaceutical excipients, as it is hard to get filaments with characteristics suitable for FDM printing. Also, if you are storing filaments have many possible combinations of drugs+polymers can you store? If you are personalising medications the drugs would differ, so will the polymer it is dispersed in...personalising medications is not just about personalizing the dose, as it also involves personalizing the drug release, which can be controlled but selecting a suitable polymer. So, how many possible combinations can be stored? What about stability issues? Also, storage space? 

Response: The reviewer is right to recommend a better explanation of these points. Different strategies can be employed to modulate the drug release as well as to associate different drugs in the same drug product according to the individual patient requirements, without the need to have a stock of filaments for each possible printing condition.

Please, see additional discussion provided in the revised manuscript on pages 8 and 9.

“Different modified drug release profiles can be achieved with the manufacture of filaments from polymers with delimited solubilization characteristics (fast, slow, pH dependent, etc.)…”

“…Besides the use of polymers with characteristics that modulate the drug release from the printed medicine, the drug release kinetics can also be directed by adjusting printing parameters such as infill percentage, infill pattern or print speed. In the case of drug combination in the same dosage form, it is possible to alternate the filaments that feeding the printer in order to build the desired structure. An innovative approach to do this can be performed with the aid of the Palette 2 device (Mosaic Manufacturing Ltd, Canada), in which different filaments are combined into one that will feed the 3D printer. This filament produced from the merger of numerous fragments of several filaments,has its composition defined according to the 3D structure to be built in the printer.”

14. Line 244-246: what about simple tests of batch-to-batch quality control please discuss further.

Response: We added a paragraph to discuss new approaches to batch-to-batch quality control of printed structures. Please, see the revised manuscript on page 10.

“Moreover, new analytical alternatives have been probed in quality control of the 3D printed structures. Among them, the use of tools with viability for batch-to-batch analysis such as the near infrared spectroscopy that can perform drug content determinations with a sensitivity comparable to that of chromatographic methods [1], and the terahertz pulsed imaging, which allow acquisition of single-depth scans in a few milliseconds providing information on the microstructure of the printed devices [54]. These new analytical approaches can make a significant contribution to the safety of printed pharmaceutical preparations”.

Reviewer 2 Report

Araujo et al. have shown another great potential of 3D printing. The 3D printing of medicine sounds really novel and interesting to me. Figures including telemedicine care cycle and adaptation of the 3D printer are comprehensive. So I would like to recommend the manuscript to be published after addressing my concerns. 

Authors need to compare in details the 3D printing and traditional production of medicines such as speed. I am concerned whether the 3D printing can be commercialized if the speed is slow. 

During 3D printing, how can medicines maintain stability or quality? In other words, will 3D printing process have any influence on the quality of medicines? 

Why is it so necessary to have different shapes of medicine? 

Author Response

Araujo et al. have shown another great potential of 3D printing. The 3D printing of medicine sounds really novel and interesting to me. Figures including telemedicine care cycle and adaptation of the 3D printer are comprehensive. So I would like to recommend the manuscript to be published after addressing my concerns. 

Response: We appreciated the careful evaluation of our manuscript provided by you. The responses to your comments are listed and addressed one-by-one below.

1. Authors need to compare in details the 3D printing and traditional production of medicines such as speed. I am concerned whether the 3D printing can be commercialized if the speed is slow. 

Response: Thank you for the suggestion. We improved the discussion on this topic on page 2 of the revised manuscript.

“The FDM 3D printer may not be the optimal solution for large-scale production. For instance, a tablet machine can be 60 times faster than a printer [15]. 3D printers cannot match the industrial tablet machines velocity, but they certainly serve to address an existing therapeutic gap with regard to the need for individualization of drug therapy, acting in a complementary or alternative manner to the conventional drug product production [16,17].  

On the other hand, in a small scale at compounding pharmacies, the speed difference between manually encapsulating powders and using a printer may not be so discrepant. The printer can also be adapted with multiple heads, making it possible to print several units at the same time, accelerating the process.”

2. During 3D printing, how can medicines maintain stability or quality? In other words, will 3D printing process have any influence on the quality of medicines?

Response: Information about quality control and stability of 3D printed devices were inserted throughout the manuscript. Please, see the revised manuscript on pages 6 and 9. 

“The thermal processing of the formulation involved in the FDM 3D printing and its risks to the drug stability cannot be ignored. In fact, stability issues for thermosensitive drugs has already been warned by the initial works using this technology [51]. The problem has been addressed by using polymers with low glass transition temperature or by using plasticizers which can reduce the polymers glass transition and consequently the nozzle temperature [34,52].”

 “The quality control required for finished drug products would be the same as those currently required for common solid preparations in compounding pharmacies, which includes average weight and organoleptic characteristics. However, due to the high automation of the 3D printing process and the small number of production steps, there is potentially a lower risk of human error, and consequently a noticeable gain in the safety of printed drug products. Extra tests to determine the drug delivery device characteristics, such as hardness, friability, drug content and drug release could be applied to pilot batches or by sampling in accordance with each country regulatory demand [25, 26, 55].”

3. Why is it so necessary to have different shapes of medicine?

Response: According to some studies, innovative geometries of tablets can increase patient compliance. Besides the aesthetic aspect, tablet shape can be used as strategy to modulate the drug-released rate, which is dependent on surface area/ volume of the tablet. This aspect is discussed in the revised manuscript on page 4.

“Innovative geometries to increase patient compliance could also be created. Indeed, tests performed in vivo revealed patients were open to new geometries, such as donut form (torus) [6]. Pediatric dosage forms imitating candy may improve children acceptance for oral forms and the extrusion and printing processes using polymers help to mask bitter active pharmaceutical ingredient flavor [36]. Besides the aesthetic aspect, control of tablet shape can be used to modulate the drug-released rate, which is dependent on surface area/ volume of the tablet [7]. Figure 3 shows several different shapes of FDM printed tablets and capsules.”

Round  2

Reviewer 1 Report

all the comments have been taken on board and the manuscript revised appropriately. it is a good review article.